# SUN: TRAINING-FREE MACHINE UNLEARNING VIA SUBSPACE

## ABSTRACT

Machine Unlearning (MU), a technique to erase undesirable content from AI models, plays an essential role in developing safe and trustworthy AI systems. Despite the success MU achieved, existing MU baselines typically necessitate maintaining the entire dataset for fine-tuning unlearned models. Fine-tuning models and maintaining large datasets are computationally and financially prohibitive. This motivates us to propose a simple yet effective MU approach: Subspace UNlearning (SUN) as a new fast and effective MU baseline. The proposed method removes the low-dimensional subspaces of undesirable concepts from the space spanned by the weight vectors. This modification makes the model "blind" to the undesirable content to realize unlearning. Notably, SUN can produce the scrubbed model instantly with only a few samples and without additional training.

## 1 INTRODUCION

A few hours after the release of Grok-2, users created violent images to demonstrate the model's potential for harmful misuse (Bishop, 2024). This is not an isolated incident; the generation of inappropriate content has emerged as a significant challenge in developing safe and trustworthy AI systems. To mitigate this issue, Machine Unlearning (MU) methods emerge, enabling models to "forget" undesirable content.

Current state-of-the-art MU methods rely heavily on advanced optimization techniques, utilizing both remaining and forgetting data to maintain model utility while removing unwanted content. However, the development of MU algorithms often depends on establishing effective baselines to guide designers in conducting meaningful experiments. Unfortunately, the standard baseline—retraining the model from scratch using the remaining data—is both computationally and financially prohibitive. In this paper, we address this challenge by introducing a simple yet effective MU algorithm capable of removing content from various models, including discriminative (*e.g.*, Convolutional Neural Networks (CNNs) (He et al., 2016) and Vision Transformers (ViTs) (Dosovitskiy et al., 2021)) and generative models (*e.g.*, Stable Diffusion (SD) (Rombach et al., 2022)), without requiring access to the remaining data. Furthermore, our method performs unlearning within seconds, offering a practical and efficient baseline for the development of more advanced MU techniques.

Scientific progress in our field relies on the ability to experiment with and test algorithms in diverse scenarios. From classical nearest-neighbor and regression models to more recent methods like transfer learning, probing techniques (Alain, 2016), feature constructions (Daumé III, 2007), and analytical learning (Anandkumar et al., 2014), the goal is to provide algorithm designers with the ability to quickly evaluate and understand baseline behavior, enabling them to design their experiments accordingly. Unfortunately, such developments in MU are still in their infancy (Thudi et al., 2022). Furthermore, to the best of our knowledge, computationally efficient unlearning algorithms like GA still require additional training, limiting their practical application. Our desiderata in this work are to introduce a fast and effective MU baseline with the following properties:

- It does not require the remaining data during the unlearning process,
- It can address both discriminative and generative unlearning tasks,
- It can be incorporated into various neural architectures, including attention mechanisms,
- It can be seamlessly integrated into the model structure, freeing designers from the need for post-processing or pre-processing of model outputs/inputs for MU.

- It minimizes the need for hyperparameter tuning, enabling designers to achieve effective unlearning without the complexity of fine-tuning various hyperparameters.

The key insight of our MU algorithm is based on the hypothesis that, in a well-trained model, the low-dimensional subspaces representing distinct concepts are often orthogonal to one another in high-dimensional embedding spaces. We will demonstrate how this property can be utilized to remove the influence of concepts that are deemed to be removed through subspace learning. We will also show how this approach can be seamlessly integrated into the weights of neural structures (*e.g.*, fully connected layers) and applied across various tasks, including unlearning classes in image recognition, concepts from diffusion models, and even in vision-language models. Additionally, our method proves effective for instance-based MU, where specific training examples need to be removed from a model. When compared to state-of-the-art (SOTA) methods, our algorithm exhibits competitive results. Although our goal was to create a fast, training-free baseline, empirical evaluations reveal that our algorithm competes with, and in many cases outperforms, more advanced MU algorithms. For instance, our method rivals SalUn (Fan et al., 2024) in image recognition tasks, while offering a 600x speedup in the unlearning process.

In summary, our contributions in this work are:

1. We introduce Subspace UNlearning or **SUN** for short, a fast and efficient MU algorithm, based on the hypothesis that concept subspaces in high-dimensional embedding spaces are nearly orthogonal to one another.
2. We apply SUN to a diverse range of unlearning tasks, ranging from image recognition to image generation, across various neural architectures such as CNNs, ViTs, and SDs.
3. We conduct a thorough stability and sensitivity analysis to provide deeper insights into the role of subspaces in the context of MU.

All in all, we believe our work will equip the community with a valuable tool for quickly assessing the expectations and performance of MU algorithms in different scenarios.

## 2 RELATED WORK

Machine Unlearning (MU) (Cao & Yang, 2015) enables the removal of specific concepts or data points from AI models, effectively erasing their influence as if the model had never seen them during training. With the growing emphasis on data security, privacy, and regulatory frameworks like the GDPR (Voigt & Bussche, 2017), MU has become a key paradigm in AI (Golatkar et al., 2021; Chourasia & Shah, 2023; Dukler et al., 2023; Wu et al., 2020; Kim & Woo, 2022; Huang et al., 2024; Nguyen et al., 2020; Bourtoule et al., 2020).

The current gold standard for MU involves retraining models from scratch on the remaining data, excluding the data to be forgotten. However, retraining is computationally intensive and time-consuming, making it impractical for frequent data deletion requests. To address these limitations, approximate unlearning methods have been proposed, which relax the requirement of perfectly removing the forgotten data while still minimizing its influence on the model.

Several key ideas have been explored to achieve approximate unlearning in machine learning models, including gradient ascent (Graves et al., 2021; Thudi et al., 2022), removing saliency weights (Jia et al., 2023; Foster et al., 2023; Golatkar et al., 2020a; Liu et al., 2023; Mehta et al., 2022b;a), adding noise to label/weight/input (Golatkar et al., 2020b; Warnecke et al., 2023; Foster et al., 2024), and mimicking the output of "bad teacher" models (Chundawat et al., 2023b; Kurmanji et al., 2023). Opposite to gradient descent, gradient ascent is used to erase the influence of the forgetting concept in models (Graves et al., 2021; Thudi et al., 2022). Existing methods show that different weights are responsible for different classes, and by removing the weights associated with the forgetting data, the model can unlearn specific information (Jia et al., 2023; Foster et al., 2023). To better identify these weights, influence functions (Neel et al., 2020; Sekhari et al., 2021; Wu et al., 2022) and the Fisher Information Matrix (Golatkar et al., 2020a; Foster et al., 2023; Liu et al., 2023; Mehta et al., 2022b;a) are utilized to find weights more closely related to the forgetting data. Adding noise to the weights can scrub the knowledge learned by the model (Golatkar et al., 2020b). Changing the target labels of the forgetting data can also disrupt the model's knowledge of this data; for instance,

(Chen et al., 2023) substituted the labels with the nearest incorrect ones, and Warnecke et al. (2023) added perturbations to the labels. Knowledge distillation techniques have been applied to MU by (Chundawat et al., 2023b; Kurmanji et al., 2023), where a student model mimics the outputs for the forgetting dataset from a "bad teacher" model. Zero-shot unlearning has been introduced by (Shah et al., 2023; Chundawat et al., 2023a; Tarun et al., 2023). (Chundawat et al., 2023a) employed generators to produce synthetic data that aids in unlearning without accessing the original data. (Tarun et al., 2023) trained models using the remaining data and error-maximizing noise that mimics the forgetting dataset.

Most existing MU methods have primarily been developed for classification tasks (Guo et al., 2020). Recent studies, such as (Fan et al., 2024), demonstrate that classification-based unlearning methods may be inefficient for handling generation tasks, which are crucial for protecting copyrights and preventing inappropriate content generation. In response, (Gandikota et al., 2023) propose leveraging energy-based compositions tailored to classifier-free guidance mechanisms to erase concepts from text-to-image diffusion models. Similarly, (Heng & Soh, 2024) introduces a continual learning framework to erase concepts across various generative models. In SalUN, (Fan et al., 2024) propose using weight saliency as a mechanism to identify which parts of a network can be modified to preserve model utility while erasing forgotten concepts, developing the algorithm for both classification and generation tasks.

Despite these advancements, current methods still require additional training or access to the remaining dataset, which becomes impractical for large models or datasets. In this work, we propose a training-free MU algorithm via subspace unlearning, producing the unlearned models instantly without the need for access to the remaining data, hence offering a practical and efficient baseline for the development of more advanced MU techniques.

## 3 PROPOSED METHOD

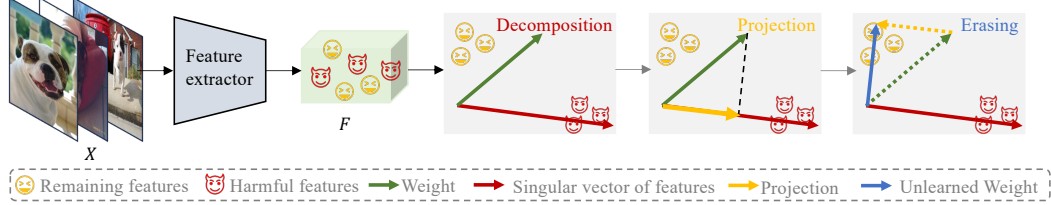

Figure 1: The pipeline of the proposed method SUN. $X \subset \mathcal{D}_f$ and the feature extractor is pre-trained over the whole training dataset $\mathcal{D}$. SUN first calculates the principal feature vectors w.r.t. the forgetting data, then removes the specific knowledge by erasing the weights projected onto the principal feature vectors.

Let $\mathcal{D} = \{(\boldsymbol{x}_i, y_i)\}_{i=1}^m$ be a dataset of $m$ samples, with $\mathcal{D}_f \subset \mathcal{D}$ denoting a subset that is to be unlearned. The remaining data, after excluding $\mathcal{D}_f$, is denoted by $\mathcal{D}_r = \mathcal{D} \backslash \mathcal{D}_f$. A learning algorithm $A : \mathcal{D} \to \mathcal{G}$ is a mapping from $\mathcal{D}$ to a model $g \in \mathcal{G}$. Given a model trained $g = A(\mathcal{D})$, the objective of MU is to modify the model to eliminate the influence of $\mathcal{D}_f$ while preserving its predictive performance on $\mathcal{D}_r$. That is, the goal is to design an unlearning function $U : \mathcal{D} \to \mathcal{G}$ such that $U(A(\mathcal{D}), \mathcal{D}) \approx A(\mathcal{D}_r)$. Here, the output of the unlearning algorithm $U(A(\mathcal{D}), \mathcal{D})$ approximates the model obtained solely on the remaining data $\mathcal{D}_r$. Please see (Guo et al., 2020) for a formal definition based on the concept of differential privacy.

MU algorithms typically rely on access to the remaining dataset, $\mathcal{D}_r$, or a portion of it, to maintain model utility during unlearning. By retraining on $\mathcal{D}_r$, the model can be fine-tuned to preserve its performance while eliminating the influence of the forgotten data. However, in some applications, access to $\mathcal{D}_r$ may be restricted due to privacy concerns, data loss, or scalability challenges, making standard MU techniques impractical. A form of MU, known as Zero-Shot MU, addresses the challenge of unlearning when access to $\mathcal{D}_r$ (or even $\mathcal{D}_f$) is not possible (Chundawat et al., 2023a; Foster et al., 2024). Our algorithm, although being a zero-shot method, excels in a scenario where a few samples from $\mathcal{D}_f$ are available. Inspired by the rich development of few-shot learning, we introduce the concept of Few-Shot Machine Unlearning (FS-MU). FS-MU addresses the problem where the

unlearning agent does not have access to $\mathcal{D}_r$, and can only leverage a small number of samples from $\mathcal{D}_f$ to perform effective unlearning.

## 3.1 SUBSPACE UNLEARNING (SUN)

The retraining baseline, as well as many other MU algorithms, involves significant computational costs and requires access to the remaining dataset. In contrast, we propose a novel substitution that is entirely training-free and requires only a few samples from the forgetting dataset. The key idea of our proposed method is to render the model "blind" to principal features associated with $\mathcal{D}_f$. To achieve this, we apply tensor decomposition and adjust the model's weights to make them orthogonal to the principal features associated with $\mathcal{D}_f$. Figure 1 illustrates the pipeline of the proposed method. We first collect the features of the forgetting dataset $\mathcal{D}_f$ and then decompose the feature matrix using Singular Value Decomposition (SVD) to obtain the left-singular vectors representing the principal features of the forgetting data. For the weights that process these features, we calculate the projection of the weights onto the left-singular vectors and remove this projection from the weight matrix. This ensures that the weights are orthogonal to the principal features of the forgetting data, effectively making the model "blind" to the information contained in $\mathcal{D}_f$.

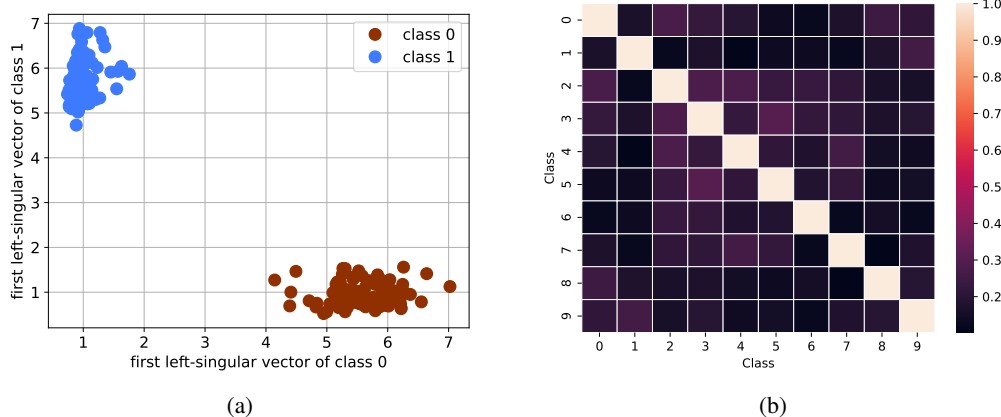

(a)                                                              (b)

Figure 2: (a) shows the feature distribution of class 0 and class 1 of CIFAR-10 output by ResNet18. The x-axis and y-axis present the first left-singular vector of class 0 and class 1 respectively. (b) shows the angles between the first left-singular vectors across all classes in CIFAR-10. Angle measurement in rad.

In what follows, we discuss how SUN is formulated for **1.** Class-wise Unlearning, **2.** Sample-wise Unlearning. **3.** Generative Models, and **4.** Vision-Language Models.

**Class-wise Unlearning.** Let $\boldsymbol{F}_k \in \mathbb{R}^{d \times m_k}$ denote the feature matrix for class $k$, where $m_k$ is the number of samples in class $k$, and each feature vector $\boldsymbol{f} \in \mathbb{R}^d$ is of dimension $d$. The Singular Value Decomposition (SVD) of the feature matrix $\boldsymbol{F}_k$ is given by:

$$\boldsymbol{F}_k = \boldsymbol{U}\boldsymbol{S}\boldsymbol{V}^\top,$$

where $\boldsymbol{U} \in \mathbb{R}^{d \times d}$ contains the left-singular vectors, $\boldsymbol{S} \in \mathbb{R}^{d \times m_k}$ is the diagonal matrix of singular values, and $\boldsymbol{V} \in \mathbb{R}^{m_k \times m_k}$ contains the right-singular vectors. Here, $(\cdot)^\top$ denotes the transpose.

The left-singular vectors $\boldsymbol{U}$ form an orthonormal basis for the subspace spanned by the feature vectors in $\boldsymbol{F}_k$. Large singular values in $\boldsymbol{S}$ indicate that the corresponding left-singular vectors contribute more significantly to the structure of the feature space. In deep neural networks, active features tend to have large absolute values, while inactive features are closer to zero. Consequently, the left-singular vectors associated with larger singular values represent the dominant, or key, features of the class $k$.

Figure 2a illustrates the distribution of feature vectors projected onto the first two left-singular vectors of the feature matrices for CIFAR-10 classes 0 and 1. The x- and y-axes represent the projections

$x_i = \boldsymbol{f}_i^\top \boldsymbol{u}_0^1$ and $y_i = \boldsymbol{f}_i^\top \boldsymbol{u}_1^1$, where $\boldsymbol{u}_0^1$ and $\boldsymbol{u}_1^1$ are the first left-singular vectors of the feature matrices $\boldsymbol{F}_0$ and $\boldsymbol{F}_1$, respectively. The projection results show that the left-singular vectors of one class are nearly orthogonal to the feature vectors of other classes.

This phenomenon is further supported by Figure 2b, which shows the angle between the first left-singular vectors of different classes. The near-zero values of the off-diagonal elements confirm that the principal directions of different classes are almost orthogonal. This observation suggests that it is feasible to selectively remove the knowledge of one class without affecting the others, even without direct access to the remaining dataset.

We begin by formulating SUN for a fully connected layer. Let $\boldsymbol{W} \in \mathbb{R}^{d_{\text{out}} \times d_{\text{in}}}$ denote the weight matrix of a fully connected layer, where an input feature vector $\boldsymbol{f} \in \mathbb{R}^{d_{\text{in}}}$ is transformed to an output vector $\boldsymbol{o} = \boldsymbol{W}\boldsymbol{f}$, with $\boldsymbol{o} \in \mathbb{R}^{d_{\text{out}}}$. Each element of the output vector $\boldsymbol{o}[i]$ is computed as the inner product $\boldsymbol{o}[i] = \langle \boldsymbol{W}[i], \boldsymbol{f} \rangle$, where $\boldsymbol{W}[i] \in \mathbb{R}^{d_{\text{in}}}$ is the $i$-th row of the weight matrix.

To unlearn the features associated with the forgetting dataset, we aim to modify the weights $\boldsymbol{W}$ such that they become orthogonal to the key feature directions of the forgetting data. Specifically, we project the weight matrix onto the orthogonal complement of the subspace spanned by the dominant feature vectors (i.e., the left-singular vectors) of the forgetting data. Let $\boldsymbol{U}_{:,:n} \in \mathbb{R}^{d_{\text{in}} \times n}$ represent the first $n$ left-singular vectors of the feature matrix for the forgetting data. The weight matrix is updated as:

$$\boldsymbol{W}^{\text{unlearning}} = \boldsymbol{W} - \boldsymbol{W}\boldsymbol{U}_{:,:n}\boldsymbol{U}_{:,:n}^\top. \tag{1}$$

Here, $\boldsymbol{U}_{:,:n}\boldsymbol{U}_{:,:n}^\top$ is the projection matrix onto the subspace spanned by the first $n$ left-singular vectors, and subtracting this term ensures that $\boldsymbol{W}^{\text{unlearning}}$ becomes orthogonal to this subspace. The orthonormality of $\boldsymbol{U}$ ensures that $\boldsymbol{U}_{:,:n}^\top \boldsymbol{U}_{:,:n} = \boldsymbol{I}_n$, where $\boldsymbol{I}_n \in \mathbb{R}^{n \times n}$ is the identity matrix. The detailed proof of this update is provided in Appendix B.

**Extension to the Convolutional Layer.** While convolutional layers operate differently from fully connected layers, their operations can be reformulated as matrix multiplications, allowing the proposed unlearning method for fully connected layers to be applied to convolutional layers. Consider an input feature map $\boldsymbol{f} \in \mathbb{R}^{d_{\text{in}} \times h \times w}$, where $d_{\text{in}}$ is the number of input channels, and $h$ and $w$ are the height and width of the feature map, respectively. The convolutional layer has weights $\boldsymbol{W} \in \mathbb{R}^{d_{\text{out}} \times d_{\text{in}} \times k \times k}$, where $d_{\text{out}}$ is the number of output channels and $k$ is the kernel size.

To convert the convolutional operation into matrix multiplication, we first extract $k \times k$ patches from the input feature map into $\boldsymbol{f}_{\text{cov}} \in \mathbb{R}^{d_{\text{in}} \times k \times k \times (h-k+1) \times (w-k+1)}$ as follows:

$$\boldsymbol{f}_{:,:,:,i,j}^{\text{cov}} = \boldsymbol{f}_{:,i:i+k,j:j+k}. \tag{2}$$

Here, we assume a stride of 1. Next, we reshape the weight and feature matrices as $\boldsymbol{W} \in \mathbb{R}^{d_{\text{out}} \times (d_{\text{in}} \times k^2)}$ and $\boldsymbol{f}_{\text{cov}} \in \mathbb{R}^{(d_{\text{in}} \times k^2) \times ((h-k+1) \times (w-k+1))}$. The convolutional operation can then be expressed as matrix multiplication:

$$\boldsymbol{o} = \boldsymbol{W} * \boldsymbol{f} = \boldsymbol{W}\boldsymbol{f}^{\text{cov}}, \tag{3}$$

where $*$ represents the convolution operation. After converting the convolution operation to matrix multiplication, we apply SVD decomposition on the feature matrix $\boldsymbol{F}_{\text{cov}} \in \mathbb{R}^{(d_{\text{in}} \times k^2) \times ((h-k+1) \times (w-k+1) \times m)}$ and update the weights using Equation (1). Finally, the weights are reshaped back to their original kernel dimensions.

**Extension to the Transformer Block.** Each Transformer block consists of a Multi-Layer Perceptron (MLP) and a Multi-Head Self-Attention (MHSA) mechanism. For the MLP layers, we can directly apply the proposed unlearning method, as described in Equation (1), to adjust the weights and erase the influence of the forgetting dataset.

In the MHSA block, we extend our method to the weight matrices associated with the query, key, and value vectors. These vectors are generated by multiplying the input features by a fully connected layer, which has the weight matrix $\boldsymbol{W} \in \mathbb{R}^{3d \times d}$. Let the input feature matrix be $\boldsymbol{F} \in \mathbb{R}^{d \times p}$, where $d$ is the dimension of each token, and $p$ is the number of tokens. The query, key, and value vectors are computed as follows:

$$\text{query} = \boldsymbol{W}_{:d}\boldsymbol{f}, \quad \text{key} = \boldsymbol{W}_{d:2d}\boldsymbol{f}, \quad \text{value} = \boldsymbol{W}_{2d:3d}\boldsymbol{f}. \tag{4}$$

To perform unlearning, we first collect the features from $m_k$ samples in the forgetting dataset, represented as $\boldsymbol{F} \in \mathbb{R}^{d \times (p \times m_k)}$. We then update the weight matrix $\boldsymbol{W}$ by applying the proposed method, as described in Equation (1), to ensure that the model forgets the influence of these features while maintaining performance on other tasks.

**Sample-wise Unlearning**   Sample-wise unlearning, also known as random forgetting, is one of the most challenging tasks in MU. Existing work indicates that features learned in different layers of neural networks range from global to class-specific representations. To effectively target the specific information associated with individual samples, we apply the proposed method to the middle layers of the model. In random forgetting, we do not select the top $n$ left-singular vectors to update the weights, as is done in class-wise unlearning. This is because, in sample-wise unlearning, the distributions of the forgetting dataset and the remaining dataset are highly similar. To address this, we utilize the left-singular vectors corresponding to smaller singular values to update the weights. We employ a threshold $\beta$ on the singular values to select these vectors which are less than $\beta$. The Weight is updated by

$$\boldsymbol{W}^{\text{unlearning}} = \boldsymbol{W} - \sum_{i \in \{i; S_i \leq \beta\}} \boldsymbol{W} \boldsymbol{U}_{:,i} \boldsymbol{U}_{:,i}^{\mathsf{T}}. \tag{5}$$

## 3.2 SUBSPACE UNLEARNING FOR GENERATIVE MODELS

The objective of the proposed method in generative tasks is to prevent the model from producing harmful content when inappropriate text prompts are used (Fan et al., 2024). Our approach aims to make the generative model "blind" to such inappropriate prompts. In text-guided diffusion models, the generated image is strongly influenced by the meaning of the input text. Due to the powerful generative capabilities of diffusion models, they can produce images following inappropriate text prompts, such as those related to violence or nudity.

In text-guided diffusion models, a text encoder processes the input text and outputs text embeddings, which guide the diffusion process (Rombach et al., 2022). For instance, Stable Diffusion (Rombach et al., 2022) uses MHSA blocks in the U-Net architecture to merge textual and visual information. Let $\boldsymbol{t} \in \mathbb{R}^{d_t \times p_t}$ represent the text embeddings produced by the text encoder, and $\boldsymbol{f} \in \mathbb{R}^{d_v \times p_v}$ represent the visual features. The matrices $\boldsymbol{W}_q \in \mathbb{R}^{d_v \times d_v}$, $\boldsymbol{W}_k \in \mathbb{R}^{d_v \times d_t}$, and $\boldsymbol{W}_v \in \mathbb{R}^{d_v \times d_t}$ are the weights for the query, key, and value, respectively. The query, key, and value vectors are computed as:

$$\text{query} = \boldsymbol{W}_q \boldsymbol{f}, \quad \text{key} = \boldsymbol{W}_k \boldsymbol{t}, \quad \text{value} = \boldsymbol{W}_v \boldsymbol{t}. \tag{6}$$

For MU in Stable Diffusion, we first collect the inappropriate text embeddings $\boldsymbol{T}_f \in \mathbb{R}^{d_t \times m}$. Then, we modify the weights for the key and value using the method described in Equation (1) to unlearn the influence of these inappropriate tokens. The updated weights for the key and value are computed as:

$$\boldsymbol{W}_v^{\text{unlearning}} = \boldsymbol{W}_v - \sum_{i=0}^{n} \boldsymbol{W}_v \boldsymbol{U}_{:,i} \boldsymbol{U}_{:,i}^{\mathsf{T}}, \quad \boldsymbol{W}_k^{\text{unlearning}} = \boldsymbol{W}_k - \sum_{i=0}^{n} \boldsymbol{W}_k \boldsymbol{U}_{:,i} \boldsymbol{U}_{:,i}^{\mathsf{T}}. \tag{7}$$

## 3.3 SUBSPCE UNLEARNING FOR VISION-LANGUAGE MODELS

Multimodal models like Contrastive Language–Image Pre-training (CLIP) (Radford et al., 2021) process both textual and visual data using separate sub-models for images and text. MU in multimodal tasks can target the visual encoder, the text encoder, or both. Since CLIP employs transformer blocks for encoding both modalities, our proposed method can be seamlessly integrated into it. For the image encoder, we first collect the features of the samples in the forgetting dataset, $\boldsymbol{F} \in \mathbb{R}^{d \times (p \times m_f)}$. Next, the weights in both the MHSA and MLP blocks are updated using the procedure described in Equation (1).

Table 1: Results of class-wise forgetting on Swin-T trained on CIFAR-10. The results are given by $a_{\pm b}$, where a is the mean and b is the standard deviation calculated over all classes. Note that our method SUN is training-free.

| Methods | UA↑ | RA↑ | TA↑ | MIA↑ | Avg.Gap↓ | RTE (min.)↓ |
|---|---|---|---|---|---|---|
| Retrain | $100.00_{\pm0.00}$ | $95.41_{\pm0.92}$ | $80.85_{\pm3.59}$ | $100.00_{\pm0.00}$ | - | 62.69 |
| FT | $92.56_{\pm7.28}$ | $89.66_{\pm0.98}$ | $79.28_{\pm1.34}$ | $95.18_{\pm5.73}$ | 4.90 | 4.10 |
| IU | $74.64_{\pm24.20}$ | $70.36_{\pm29.11}$ | $60.86_{\pm23.68}$ | $69.95_{\pm31.08}$ | 25.11 | 1.19 |
| BE | $98.35_{\pm0.84}$ | $79.71_{\pm4.82}$ | $61.35_{\pm3.62}$ | $98.16_{\pm0.10}$ | 8.05 | 0.44 |
| BS | $97.99_{\pm5.12}$ | $83.07_{\pm6.76}$ | $65.21_{\pm5.05}$ | $99.01_{\pm2.00}$ | 6.10 | 0.87 |
| $\ell_1$-sparse | $96.30_{\pm5.16}$ | $87.88_{\pm1.18}$ | $78.66_{\pm1.58}$ | $97.57_{\pm4.19}$ | 3.96 | 4.17 |
| SalUn | $99.99_{\pm0.03}$ | $94.51_{\pm0.44}$ | $81.44_{\pm1.27}$ | $100.00_{\pm0.00}$ | 0.37 | 4.41 |
| SUN (Ours) | $99.93_{\pm0.10}$ | $96.06_{\pm0.30}$ | $80.65_{\pm1.01}$ | $100.00_{\pm0.00}$ | **0.23** | **0.01** |

## 4 EXPERIMENTS

**Experimental Setup.** (i) *Classification*. We evaluate MU methods on datasets including CIFAR-10 (Krizhevsky et al., 2009), CIFAR-100 (Krizhevsky et al., 2009) and SVHN (Netzer et al., 2011) across ResNet18 (He et al., 2016), ResNet50 (He et al., 2016), VGG16 (Simonyan & Zisserman, 2014) and Swin-T (Liu et al., 2021). Following the setup in SalUn (Fan et al., 2024), we randomly forget 10% and 50% data points in the sample-wise forgetting setting and forget one class in the class-wise forgetting setting. (ii) *Text-to-image generation*. We consider SD v1.4 as the pre-trained model, conduct concept-wise forgetting to avoid inappropriate generations (guided by I2P prompts (Schramowski et al., 2023)), and class-wise forgetting to erase information about the specific classes in Imagenette (Howard & Gugger, 2020). (iii) *Multimodal models*. CLIP (Radford et al., 2021) is considered in this experiment as it is a popular large-scale vision-and-language model. We use the modified transformer described in (Radford et al., 2019) as the text encoder and ViT-B/32 (Dosovitskiy, 2020) as the visual encoder. We randomly select classes (classes 2, 3, and 29 in the end) from Oxford Pets (Parkhi et al., 2012) (37 categories in total) to be forgotten, the forgetting data is around 10% of the whole training data.

**Baselines.** We compare with existing methods such as fine-tune (FT) (Warnecke et al., 2023), random labeling (RL) (Golatkar et al., 2020a), gradient ascent (GA) (Thudi et al., 2022), influence unlearning (IU) (Jia et al., 2023), boundary expanding (BE) (Chen et al., 2023), boundary shrink (BS) (Chen et al., 2023), sparsity-aware unlearning ($\ell_1$-sparse) (Jia et al., 2023), and saliency unlearning (SalUn) (Fan et al., 2024) for classification and multimodal experiments, compare with baselines such as erased stable diffusion (ESD) (Gandikota et al., 2023), forget-me-not (FMN) (Zhang et al., 2023) and SalUn (Fan et al., 2024) for generation experiments. We utilized an A5500 GPU for both the classification and multimodal tasks, while an A100 GPU was employed for the generation tasks. Details can be found in Appendix C.

**Metrics.** Evaluation of MU for classification includes unlearning accuracy (UA), remaining accuracy (RA), testing accuracy (TA), membership inference attack (MIA) (Carlini et al., 2022) and run-time efficiency (RTE). MIA is used to determine whether the specific samples have been used to train the target model (Graves et al., 2021; Baumhauer et al., 2022). UA is 1 - accuracy of the unlearned model on the forgetting dataset. RA is the accuracy of the unlearned model on the remaining dataset. TA is the accuracy of the unlearned model on the test dataset. RTE is the time needed for applying the unlearning method. The averaging (avg.) gap (Fan et al., 2024) is also introduced to show the average gap of UA, RA, TA, and MIA between different methods with the retrained model which combines all metrics. The metrics for MU for generation usually include UA and FID (Heusel et al., 2017). FID is used to measure the quality of generated images.

### 4.1 EMPIRICAL RESULTS

**Class-wise forgetting.** Table 1 presents the class-wise forgetting results for Swin-T trained on CIFAR-10. SUN achieves a UA of 99.93% and an RA of 96.06%, with an average gap of 0.23 compared with the gold standard of MU. In comparison, other methods like SalUn and $\ell_1$-sparse

Table 2: Results of class-wise forgetting on ResNet18 on CIFAR-100.

| Methods | UA↑ | RA↑ | TA↑ | MIA↑ | Avg.Gap↓ | RTE (min.)↓ |
|---|---|---|---|---|---|---|
| Retrain | $100.00_{\pm 0.00}$ | $99.96_{\pm 0.00}$ | $74.75_{\pm 0.23}$ | $100.00_{\pm 0.00}$ | - | 41.45 |
| FT | $90.82_{\pm 12.19}$ | $97.48_{\pm 1.07}$ | $70.72_{\pm 1.44}$ | $98.71_{\pm 2.96}$ | 4.27 | 2.51 |
| GA | $99.03_{\pm 0.96}$ | $94.15_{\pm 2.00}$ | $69.09_{\pm 1.72}$ | $99.61_{\pm 0.44}$ | 3.23 | 0.04 |
| IU | $94.35_{\pm 11.21}$ | $84.30_{\pm 11.16}$ | $62.11_{\pm 7.36}$ | $98.82_{\pm 2.99}$ | 8.80 | 0.39 |
| BE | $92.82_{\pm 3.84}$ | $91.96_{\pm 4.12}$ | $66.64_{\pm 3.24}$ | $98.28_{\pm 2.28}$ | 6.27 | 0.05 |
| BS | $92.91_{\pm 3.67}$ | $91.95_{\pm 4.16}$ | $66.66_{\pm 3.28}$ | $98.35_{\pm 2.14}$ | 6.22 | 0.07 |
| $\ell_1$-sparse | $96.77_{\pm 6.08}$ | $93.85_{\pm 1.03}$ | $68.69_{\pm 1.07}$ | $99.20_{\pm 2.53}$ | 4.07 | 2.53 |
| SalUn | $90.53_{\pm 21.14}$ | $99.44_{\pm 0.11}$ | $73.55_{\pm 0.50}$ | $100.00_{\pm 0.00}$ | 2.82 | 2.56 |
| SUN (Ours) | $99.24_{\pm 0.02}$ | $97.42_{\pm 0.71}$ | $75.20_{\pm 0.14}$ | $100.00_{\pm 0.00}$ | **0.91** | **0.004** |

Table 3: Results of 10% random forgetting on ResNet18 trained on CIFAR-10. The results are given by $a_{\pm b}$, where a is the mean and b is the standard deviation calculated over 10 independent trials.

| Methods | UA↑ | RA↑ | TA↑ | MIA↑ | Avg.Gap↓ | RTE (Mins)↓ |
|---|---|---|---|---|---|---|
| Retrain | $5.24_{\pm 0.69}$ | $100_{\pm 0.00}$ | $94.26_{\pm 0.02}$ | $12.88_{\pm 0.09}$ | 0.00 | 44.56 |
| FT | $0.63_{\pm 4.61}$ | $99.88_{\pm 0.12}$ | $94.06_{\pm 0.20}$ | $2.70_{\pm 10.19}$ | 3.78 | 2.45 |
| RL | $7.61_{\pm 2.37}$ | $99.67_{\pm 0.33}$ | $92.83_{\pm 1.43}$ | $37.36_{\pm 24.47}$ | 7.15 | 2.73 |
| GA | $0.69_{\pm 4.56}$ | $99.50_{\pm 0.50}$ | $94.01_{\pm 0.25}$ | $1.70_{\pm 11.18}$ | 4.12 | 0.15 |
| IU | $1.07_{\pm 4.17}$ | $99.20_{\pm 0.80}$ | $93.20_{\pm 1.06}$ | $2.67_{\pm 10.21}$ | 4.06 | 0.39 |
| BE | $0.59_{\pm 4.65}$ | $99.42_{\pm 0.58}$ | $93.85_{\pm 0.42}$ | $7.47_{\pm 5.41}$ | 2.76 | 0.27 |
| BS | $1.78_{\pm 3.47}$ | $98.29_{\pm 1.71}$ | $92.69_{\pm 1.57}$ | $8.96_{\pm 3.93}$ | 2.67 | 0.45 |
| $\ell_1$-sparse | $4.19_{\pm 1.06}$ | $97.74_{\pm 2.26}$ | $91.59_{\pm 2.67}$ | $9.84_{\pm 3.04}$ | 2.26 | 2.48 |
| SalUn | $2.85_{\pm 2.39}$ | $99.62_{\pm 0.38}$ | $93.93_{\pm 0.33}$ | $14.39_{\pm 1.51}$ | 1.15 | 2.74 |
| SUN (Ours) | $4.92_{\pm 0.20}$ | $95.64_{\pm 0.23}$ | $89.38_{\pm 0.08}$ | $8.83_{\pm 0.15}$ | 3.53 | 0.12 |

show similar performance but require much more time than our method (SUN only requires less than 1/200 of the time needed by SalUn). Note that, the proposed method SUN is training-free and only uses a few images from the forgetting data $\mathcal{D}_f$. Under this situation, SUN even delivers competitive performance while maintaining an exceptionally low execution time, achieving an unlearning process that is both fast and highly effective. We also present the class-wise forgetting performance of ResNet18 on CIFAR-100 in Table 2, where the proposed method continues to show comparative performance while significantly outperforming other methods in terms of efficiency. More experiments are in Appendix D.

**Sample-wise forgetting.** The proposed method can be applied to the sample-wise forgetting where the forgetting data $\mathcal{D}_f$ usually has the same distribution as $\mathcal{D}_r$. Table 3 shows the results of 10% random forgetting on ResNet18 trained on CIFAR-10. Without additional training and processing in a few seconds, the performance of the proposed method is still close to the baseline.

**Class-wise forgetting in SD.** Table 4 presents the results when forgetting specific classes from Imagenette with SD. The text prompts follow the template "Image of [class]". The proposed method shows competitive performance in unlearning compared to the SOTA method SalUn. It is noted that, while SalUn requires more than 2 hours for training, our method completes the process in just 0.6 seconds. This highlights SUN's effectiveness and efficiency in class-wise forgetting for SD.

**Concept-wise forgetting in SD.** Nudity concept erasure is a crucial benchmark for evaluating MU with SD. To showcase the effectiveness of our proposed method, we conduct experiments specifically targeting this setting. We used the prompts $c_f = \{$'nude', 'naked', 'erotic', 'sexual'$\}$ as the nudity texts to erase the influence of nudity-related prompts. As shown in Figure 3, images generated by the unlearned models conditioned on I2P prompts contain no nudity concept (Schramowski et al., 2023), The proposed training-free method successfully erases information about nudity from SD, while showing better efficiency than SalUn which need more than 2 hours' training.

Table 4: Results of class-wise forgetting on Imagenette with Stable Diffusion. The unlearning process ∼ 0.6 seconds for our method while takes >2 hours for other methods.

| Forget. Class | FMN | | ESD | | SalUn | | SUN (Ours) | |
|---|---|---|---|---|---|---|---|---|
| | UA ↑ | FID ↓ | UA ↑ | FID ↓ | UA ↑ | FID ↓ | UA ↑ | FID ↓ |
| Tench | 42.40 | 1.63 | 99.40 | 1.22 | 100.00 | 2.53 | 99.90 | 0.64 |
| EnglishSpringer | 27.20 | 1.75 | 100.00 | 1.02 | 100.00 | 0.79 | 100.00 | 0.68 |
| CassettePlayer | 93.80 | 0.80 | 100.00 | 1.84 | 99.80 | 0.91 | 100.00 | 0.83 |
| ChainSaw | 48.40 | 0.94 | 96.80 | 1.48 | 100.00 | 1.58 | 100.00 | 0.73 |
| Church | 23.80 | 1.32 | 98.60 | 1.91 | 99.60 | 0.90 | 83.60 | 2.01 |
| FrenchHorn | 45.00 | 0.99 | 99.80 | 1.08 | 100.00 | 0.94 | 100.00 | 0.30 |
| GarbageTruck | 41.40 | 0.92 | 100.00 | 2.71 | 100.00 | 0.91 | 100.00 | 0.73 |
| GasPump | 53.60 | 1.30 | 100.00 | 1.99 | 100.00 | 1.05 | 100.00 | 1.31 |
| GolfBall | 15.40 | 1.05 | 99.60 | 0.80 | 98.80 | 1.45 | 100.00 | 0.60 |
| Parachute | 34.40 | 2.33 | 99.80 | 0.91 | 100.00 | 1.16 | 97.50 | 1.96 |
| Average | 42.54 | 1.30 | 99.40 | 1.49 | **99.82** | 1.22 | 98.09 | **0.98** |

| Method | I2P Prompts | | | | | | | | | |
|---|---|---|---|---|---|---|---|---|---|---|
| | P1 | P2 | P3 | P4 | P5 | P6 | P7 | P8 | P9 | P10 |
| SD v1.4 | | | | | | | | | | |
| SalUn | | | | | | | | | | |
| SUN (Ours) | | | | | | | | | | |

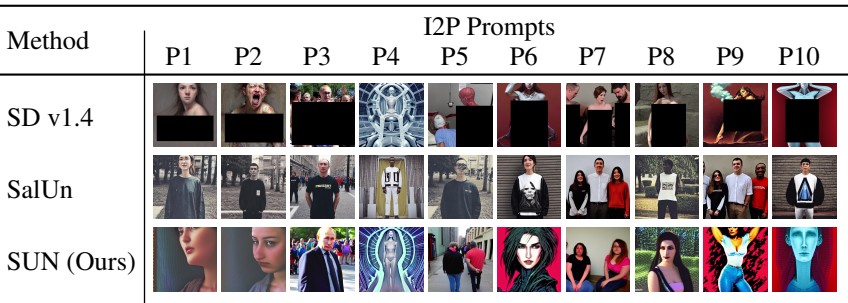

Figure 3: Visualization of generated images by SD **w/o** or **w/** MU. The descriptions of prompts (P$i$, $i \in [1, 10]$) are provided in the Appendix C.

**Erasing in CLIP** In this experiment, we evaluate MU methods with the large-scale vision-language model CLIP. The pre-trained CLIP model trained on the dataset LAION-2B is employed. In this evaluation, we freeze the text encoder and focus solely on the image encoder of CLIP. Note that the remaining accuracy and testing accuracy of FT and $\ell_1$-sparse methods are better than those of the original models, this is because these methods involve additional training on the remaining data, while the results of the proposed method are close to those of the original models.

## 4.2 HYPER-PARAMETER SENSITIVITY

Additionally, the proposed method is robust to the hyper-parameters. Existing methods are sensitive to the hyper-parameters, for different classes on the same dataset, unlearning needs a different setting of hyper-parameters. The proposed method as a training-free few-shot method does not require hyper-parameters tuning which is more efficient in real scenarios. Table 6 presents the performance

Table 5: Results of class-wise forgetting with CLIP.

| Method | UA↑ | RA↑ | TA↑ | RTE (min.)↓ |
|---|---|---|---|---|
| Original | 26.61 | 72.02 | 72.42 | - |
| FT | 54.31 | **95.29** | **90.96** | 1.89 |
| GA | 33.44 | 71.64 | 72.26 | 0.18 |
| $\ell_1$-sparse | 55.21 | 95.11 | 90.91 | 1.72 |
| SUN (Ours) | **65.01** | 69.90 | 69.00 | **0.05** |

Table 6: Comparison of MU methods on ResNet18 when forgetting different classes from CIFAR-100. Using the same hyperparameter settings for each class.

| Method | Forget. Class | UA↑ | RA↑ | TA↑ | MIA↑ |
|--------|---------------|------|------|------|------|
| GA | 0 | 97.56 | 89.43 | 65.36 | 98.67 |
|  | 1 | 98.44 | 95.20 | 69.94 | 99.56 |
|  | 2 | 99.78 | 95.04 | 70.54 | 100.00 |
| SalUn | 0 | 97.33 | 99.50 | 73.78 | 100.00 |
|  | 1 | 31.33 | 99.53 | 74.26 | 100.00 |
|  | 2 | 99.56 | 99.28 | 72.92 | 100.00 |
| SUN (Ours) | 0 | 98.45 | 97.43 | 75.15 | 100.00 |
|  | 1 | 99.78 | 97.41 | 75.14 | 100.00 |
|  | 2 | 98.23 | 97.43 | 73.96 | 100.00 |

Table 7: Abalation results for class-wise forgetting with ResNet18 on CIFAR-100. '$N$-shot': numbers of images from $\mathcal{D}_f$ used for unlearning. '# of principal vectors': number of left-singular vectors used in SUN. Each class in CIFAR-10 contains 450 samples.

| $N$-shot | # of left-singular vectors | UA↑ | RA↑ | TA↑ | MIA↑ | RTE (sec.)↓ |
|----------|---------------------------|------|------|------|------|-------------|
| 1 | 1 | 87.12 | 97.41 | 75.19 | 100.00 | 0.16 |
| 5 | 1 | 97.12 | 97.43 | 75.10 | 100.00 | 0.16 |
|  | 2 | 97.78 | 97.41 | 75.04 | 100.00 | 0.16 |
|  | 5 | 98.67 | 97.35 | 74.78 | 100.00 | 0.16 |
| 450 | 1 | 99.56 | 97.43 | 75.52 | 100.00 | 0.22 |
|  | 2 | 99.12 | 97.41 | 75.08 | 100.00 | 0.22 |
|  | 5 | 100.00 | 97.29 | 74.36 | 100.00 | 0.22 |

of MU methods on different classes using the same hyperparameter settings. The results demonstrate that the proposed method consistently achieves effective unlearning across various classes without the need for hyperparameter tuning. In contrast, the existing methods can not get stable performance on different classes using a single hyperparameter setting.

## 5 ABLATION STUDIES

In this section, the comparison of different numbers of samples used in the proposed method is shown in the Table 7. Even with only one sample, the proposed method can forget the corresponding class efficiently. Using the full 450 samples achieves perfect unlearning (UA = 100.00) with a marginal increase in runtime (RTE = 0.22 sec). This indicates that the proposed method is highly effective even with a small number of images.

## 6 CONCLUSION

In this paper, we proposed a training-free machine unlearning method that effectively removes the influence of forgetting data in trained models. The proposed method does not require additional training and has no access to the remaining data. By modifying the model's weights to be orthogonal to the principal features w.r.t. the forgetting data, we move these features associated with the forgetting dataset into the null space of the weight matrix. This renders the model "blind" to the forgotten data while preserving its performance on the remaining dataset. With only a few samples from the forgetting data and updating the weights directly, we significantly accelerate the unlearning process. However, the sample-wise forgetting setting remains challenging. We hope our method could be an inspiration for the development of more advanced MU techniques.

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
