# OpenReview forum: "SUN: Training-free Machine Unlearning via Subspace"
_ICLR.cc/2025/Conference — ICLR 2025 Conference Withdrawn Submission_

### Official Review · Reviewer_czA6 · 2024-10-27

**Soundness:** 2
**Presentation:** 3
**Contribution:** 2
**Rating:** 5
**Confidence:** 4

**Summary:**

The paper proposes to an unlearning algorithm by modifying weights to align with the orthogonal direction of undesired features.  By subtracting the projected weights against the left singular space of the unwanted features, the models can forget about those features. This method requires no training and backpropagation, which is tbh e main selling point.

**Strengths:**

* Simple and straightforward idea
* Presentation is clear and easy to follow
* Experiments show promising results

**Weaknesses:**

* in the SD experiment where concept of “nudity” is removed, there is no quantitative comparison against the baseline. I would like to see concrete evidence that removing the concept of “nudity” doesn’t impact the quality of generating humans.
* The potential entanglement of features is not discussed.
* Although the paper claims to be motivated by adversarial and unwanted behaviors of generated models, the majority of experiments are non generative, such as CLIP and Cifar-10 classifier. It would be way much more convincing if more generative models are experimented with, such as LLMs.
* This method is highly similar to the line of model merging and weight interpolation works in NLP research. Some more citations and discussions of related works is highly recommended.

**Questions:**

* How are the magnitudes of the activations preserved, especially for the layers with biases? Or maybe it doesn’t matter in practice?

* How to guarantee the removed features are fully disentangled with some of the useful features?

* Does the choice of removal data matter?  Is it better to use real data or data samples from the model itself?

**Details Of Ethics Concerns:**

When removing features with this method, there is no guarantee that some features that are useful in serving specific group are not eliminated along the way, which could create another form of bias and discrimination. This concern is not addressed or identified by the authors.

---

### Official Review · Reviewer_VDJi · 2024-10-28

**Soundness:** 2
**Presentation:** 2
**Contribution:** 2
**Rating:** 3
**Confidence:** 4

**Summary:**

Machine unlearning aims to unlearn the undesirable knowledge from the trained model, which is pivotal for trustworthy and safe AI system. To this end, existing unlearning algorithms requires the whole dataset including the remaining samples as well as the forget samples, which hinder efficient unlearning. For an effective unlearning baseline, the authors proposed SUN, which modifies the weight vectors to make orthogonal to the principal subspaces of forget concepts.

**Strengths:**

The authors provide experimental results on both classification and generative tasks.

**Weaknesses:**

- The motivation and justification of the proposed method is weak. The authors present Figure 2 to support the hypothesis, but it does not ensure that orthogonalization is crucial for unlearning. I think the authors should visualize how the unseen samples behave on the feature space. Also, could the authors provide the same visualization on sample-wise unlearning with details?
- Why the authors conduct SVD on the feature matrix, instead of the feature covariance matrix (ex. FF^T)?
- The strategy for sample-wise unlearning is too heuristic. Why should we have different strategies for class-wise and sample-wise unlearning? (actually both tasks are just to remove the knowledge of certain samples) Because the main point of this paper is that we don’t need to have the remaining samples, such having the pre-knowledge about the task (whether it is class-wise or sample-wise) is a kind of cheating.
- Need more comprehensive settings, such as (1) subclass forgetting in CIFAR-20, which aims to remove a subclass of superclass, (2) multi-class forgetting
- No details about reproducing. How to reproduce other baselines, such as FT, GA, and so on? Early stopping affects the results significantly in unlearning baselines, so it is crucial to specify how to select the final model for each baseline algorithm.
- Quantitative results for concept-wise forgetting is needed in Figure 3, such as CLIP or nudity score.

**Questions:**

- Results of original model should be included in all tables.
- Refer to Table 5 is missing in L464-469.
- Ablative results for random forgetting is needed, which is harder setting than class-wise forgetting so it will present the efficacy more effectively.

---

### Official Review · Reviewer_rJED · 2024-11-03

**Soundness:** 2
**Presentation:** 3
**Contribution:** 2
**Rating:** 3
**Confidence:** 4

**Summary:**

This paper proposes a novel machine unlearning method, SUN, to erase specific data from pretrain models without the remaining data.
Moreover, SUN can handle different unlearning scenarios including classification, generation and multimodal tasks. SUN is designed based  on the hypothesis that the features of different samples is orthogonal in high-dimension feature space. Experiments show that SUN can obtain comparable performance while requiring much less computational resources without the remaining data.

**Strengths:**

1. SUN is remaining data free, and even needs few forgetting data, which makes SUN practical in real-world applications.
2. SUN can handle multiple unlearning scenarios, including classification, generation and multimodal tasks.
3. The paper is well-written.

**Weaknesses:**

1. False Unlearning. MU aims to remove the influence of specific data from the whole model, including the feature extractor and feature processor (for example, the feature extractor). However, it seems that SUN does not modify the feature extractor.

	In **class-wise unlearning**, according to the neural collapse theory, the features of different class and the fully connected layer's weight would converge to a simplex. The projection operation of SUN seems to delete the target class weight and the output would be zero.

	In **concept-wise SD unlearning**, SUN just modifies the text encoder. Does this imply that the backbone UNET can still generate the concept? Moreover, SUN uses four concept words to erase the nudity concept, which makes me doubt if the test prompts are good enough. Please see Question 2.

2. The classification experiments should be conducted on more complex datasets, including CIFAR-100 and TinyImageNet.

3. In class-wise unlearning, SUN should be evaluated on sub-class unlearning. And the authors should include more classes.

4. Since SUN does not need the remaining data, SUN should be compared to other zero-shot methods, such as [1] and [2]. More advanced methods, such as SCRUB[3] and Neggrad+(also proposed in [3], combining finetuning and gradient ascent) should also be included.

5. The different projection operations in class-wise unlearning and sample-wise unlearning are not convincing and make me confused. In class-wise unlearning, SUN uses the vectors with large singular values, but it uses small singular values. The authors claim it's because the features of forgetting data and remaining data are similar, but the explanation seems to violate the basic hypothesis  of SUN, i.e., the separability of feature space.

[1] Foster, Jack, et al. "Zero-shot machine unlearning at scale via lipschitz regularization." arXiv preprint arXiv:2402.01401 (2024).

[2] Cheng, Xinwen, Zhehao Huang, and Xiaolin Huang. "Machine Unlearning by Suppressing Sample Contribution." arXiv preprint arXiv:2402.15109 (2024).

[3] Kurmanji, Meghdad, et al. "Towards unbounded machine unlearning." Advances in neural information processing systems 36 (2024).

**Questions:**

1. Why the MIA results of class-wise unlearning and sample-wise unlearning are so different? As shown in [1] and [2], in class-wise unlearning, the forgetting data would be predicted with low confidence after unlearning, and would tend to be classified as test samples. In sample-wise, the forgetting samples tend to be classified as training samples. unlearning Can you provide the results with other MIA implementation?

2. How is model utility is perserved in SD concept-wise forgetting experiment? The authors seems only report some visual results. Can you provide more quantitive results to eval the forgetting effectiveness and model utility?

3. Are there any results where the projection operation is conducted on convolutional layer, as you describe the extension to convolutional layer?

---

### Official Review · Reviewer_igvY · 2024-11-03

**Soundness:** 3
**Presentation:** 2
**Contribution:** 2
**Rating:** 5
**Confidence:** 4

**Summary:**

This paper proposes a training-free unlearning approach that is designed to remove the influence of a given forget set without applying gradient-based updates and without requiring access to the remainder of the dataset. Specifically, the proposed approach, given a matrix of trained weights, leverages tensor decomposition to find the principal directions associated with the forget set. They then modify the weights to make them orthogonal to those directions (by subtracting from the weights the projection of the weights onto the left-singular vectors with the largest singular values obtained by a SVD decomposition of the forget set features). They propose instantiations of this method for various unlearning problem formulations: class unlearning, instance unlearning, and concept erasure in vision and vision-language models. They show empirically that the proposed method is competitive with prior work while being much faster for some class unlearning and concept erasure applications, while performing worse than other methods in terms of utility drop in instance-wise unlearning. The authors also conduct analyses and show that this method is more robust to hyperparameters compared to prior work and can enable concept unlearning given only a small number of examples.

**Strengths:**

- The approach is agnostic to the architecture and model and can be instantiated for different unlearning problems in discriminative models, as well as for unlearning in generative models for text-to-image generation and vision-language models.

- Being training-free, this method is extremely fast compared to other methods. In several cases / applications, it is shown to achieve comparable results to SalUn while being drastically faster.

- The demonstrated robustness to hyperparameters is an important strength of this method and a common drawback of other unlearning methods.

- The proposed method is able to remove a class given only a few samples from it, which is desirable in some scenarios.

**Weaknesses:**

- Some important related work is missing. In particular, it seems that the proposed approach is very similar (training-free, use of SVD) to that of [1] in the references below. Could the authors comment on the similarities and differences from that work?

- For the application of mitigating generation of harmful content, it seems appropriate to also compare to “alignment” methods, e.g. [2] in the references below, since, despite having different terminology, it seems that the two are aimed at addressing the same problem.

- Introduction and related work. The narrative in Sections 1 and 2 is a little confusing. The authors claim that state-of-the-art methods require the retain set, but then in the related work they correctly acknowledge the existence of zero-shot methods that don’t require the retain set. Doesn’t that refute the previous claim? Also, it would have been useful to compare against those methods empirically. It would be great to have a more holistic discussion about the goal of this work and the strengths and weaknesses of existing methods (including all those that are mentioned in the related work) in addressing those goals.

- It seems that the proposed method is nicely motivated in the case where we are interested in class unlearning, and in particular when the class that we want to unlearn has left singular vectors that are (nearly) orthogonal to those of classes that we want to retain. Figure 2 clearly shows that this is the case for classes in CIFAR-10. However, it is not clear how this approach would perform in situations where the class to forget is very “similar” to some of the retain classes, e.g. in a more complex fine-grained classification task.

- Further, for instance-wise unlearning, is there some empirical evidence that example-based information is stored in the directions associated with singular vectors of “medium” values, to justify the proposed approach in this case? It is not obvious to me why this would be the case, and it seems that, depending on the chosen threshold value, the proposed approach would end up unlearning “too much” around the given forget set (note that the poorer performance in Table 3 for SUN compared to other methods corroborates this hypothesis). It seems that this method is by design not well-suited for instance-class unlearning. Indeed, the way that the method is motivated in the introduction is that it is “based on the hypothesis that concept subspaces in high-dimensional embedding spaces are nearly orthogonal to one another” – which does not speak to how to remove the influence of specific instances from a model. It would be great to add more discussion around this in the paper.

- Building on the above point, the discussion of how the proposed method “requires only a few samples from the forgetting dataset” makes it seem ill-suited for instance-wise unlearning, where the goal is to forget only the given instances, not entire concepts indexed by those. It would be great to add some discussion of these differences and limitations. It seems cleaner to me overall to limit the scope here to removal of entire classes or concepts (in discriminative or generative models); those are also the cases where SUN performs better empirically, unsurprisingly.

- Some factual errors in the related work. SCRUB (Kurmanji et al) does not have a bad teacher. It only has a single teacher (the original model) – the training objective is to distill from that teacher as usual for the retain set, and to modify the distillation objective for the forget set by multiplying with -1. This is different from (Chundawat et al) that employs two different teachers.


Minor:
- Figure 1 seems to be specific to harmful features?
- Line 150: the unlearning algorithm U should also take as input the original model (or a training algorithm), so U: D → G isn’t fully accurate.
- Line 151: it is more correct (based on the cited paper there, for instance) to talk about indistinguishability between distributions of models, rather than models (because a different choice of a random seed would yield different model weights).


References

[1] Deep Unlearning: Fast and Efficient Gradient-Free ClassForgetting. TMLR, 2024.

[2] Improving Alignment and Robustness with Circuit Breakers

**Questions:**

- “FID is used to measure the quality of the generated images” – is this applied on generated images for “retained concepts” or “forgotten concepts”, or both? Please clarify.

- Is the proposed approach applied to all layers of the model? Have the authors investigated the effect of which layers to apply it to?

- For the generative model applications, how is the “accuracy” of generated images measured, for the UA metric? Is an external classifier used here? If so, how was that model trained?

---

### Note · Authors · 2024-11-15

**Comment:**

Thanks to the reviewers for their work and suggestions.

**Withdrawal Confirmation:**

I have read and agree with the venue's withdrawal policy on behalf of myself and my co-authors.